# Constructing the Ecological Security Pattern of Nujiang Prefecture Based on the Framework of “Importance–Sensitivity–Connectivity”

**DOI:** 10.3390/ijerph191710869

**Published:** 2022-08-31

**Authors:** Yimin Li, Juanzhen Zhao, Jing Yuan, Peikun Ji, Xuanlun Deng, Yiming Yang

**Affiliations:** 1College of Earth Sciences, Yunnan University, Kunming 650091, China; 2College of International Rivers and Eco-Security, Yunnan University, Kunming 650091, China; 3Chongqing Institute of Surveying and Mapping, Chongqing 401120, China

**Keywords:** comprehensive evaluation of ecological security, ecological corridor, ecological node, ecological security pattern, minimum cumulative resistance model

## Abstract

Constructing an ecological security pattern is vital to guaranteeing regional ecological security. The terrain and geomorphology of the alpine valley are complex and sensitive, meaning it is difficult to construct ecological security patterns. Therefore, the study takes Nujiang Prefecture as the study area and builds an “Importance–Sensitivity–Connectivity” (Importance of ecosystem service, eco-environmental sensitivity, and landscape connectivity) framework to carry on the comprehensive evaluation of the ecological security and identification of ecological sources. Furthermore, we constructed an ecological resistance surface using land-use type. Using the minimum cumulative resistance (MCR) model, the study identifies the ecological corridors and nodes to build ecological security patterns to optimize the ecological spatial structure of Nujiang Prefecture. The results showed that (1) the importance of ecosystem services was higher in the west and lower in the east. The high-sensitive areas of the ecological environment were distributed discontinuously along the banks of the Nujiang and the Lantsang River, and the areas with high landscape connectivity were distributed in patches in the Gaoligong Mountain Nature Reserve and the Biluo Snow Mountain. (2) The overall ecological security was in a good state, and the ecologically insecure areas were primarily distributed in Lanping County and the southeast region of Lushui City. (3) The primary ecological source area was identified to be 3281.35 km^2^ and the secondary ecological source area to be 4224.64 km^2^. (4) In total, 26 primary ecological corridors, 39 secondary ecological corridors, and 82 ecological nodes were identified.

## 1. Introduction

In recent years, with the rapid development of economic and social life, the intensification of human activities has led to the over-exploitation of natural resources and serious environmental pollution, resulting in a series of ecological and environmental problems, such as water resource shortages [1,2], biodiversity loss [3,4], and soil erosion [5,6]. Ecological security, therefore, cannot be ignored. Attention to ecological security originated from land health research in 1941 [7], after which the International Institute for Applied Systems Analysis (IIASA) first proposed the concept of ecological security in 1989. Ecological security refers to the ability of an ecosystem to provide ecosystem services for human survival and economic and social development in a reasonable structure, complete functions, and stable patterns within a country or region [8,9]. As more people pay attention to the economic and social issues involved in the ecological environment, ecological security has become a hot topic in international ecosystem research and a new theme of sustainable development of human society [10]. In particular, China has realized great achievements in economic development over the past four decades. However, its high-investment, high-consumption development mode has not only consumed a tremendous amount of natural resources but also caused serious damage to the ecosystem. China’s attention to the protection of ecological security is gradually improving in this context. Ecological security has become a research hotspot in ecology [11,12,13], geography [14,15,16], environmental science [17,18], and other fields, and the concept of constructing ecological security patterns was proposed to improve ecosystem service function and lead the way for ecological security [19].

An ecological security pattern is an interconnected ecological network composed of different ecosystems. It is an effective way to support biological species, maintain natural ecological processes, improve regional ecological security, and realize ecological security [8,20]. It is additionally an important method to identify protection, strengthen connectivity, and manage effectiveness within a regional scope [21,22]. It plays a controlling role in the maintenance and protection of regional ecosystems [23]. At present, the ecological security pattern policy has become one of the important national strategies for coordinating ecosystem protection and economic development in China and has been identified as a bottom-line method for priority areas under protection [24]. Most studies have analyzed and discussed the ecological security pattern from the aspects of concept [11,25,26], technical methods [27,28,29], multi-level network construction [30,31,32], and regions with different physical geographical characteristics [33,34], thereby enriching the practical framework of the theory. At present, the main paradigms for establishing ecological security patterns are (1) identifying ecological sources, (2) constructing ecological resistance surfaces, (3) extracting ecological corridors, and (4) extracting ecological nodes.

Ecological sources are defined as priority areas that provide high-level ecosystem services. Existing studies primarily identify them in two ways. The first is the artificial determination by simply selecting woodland, nature reserves, scenic spots, and habitats of key species [35,36,37]. The second method would be a goal-oriented evaluation index system, which is used to evaluate the ecological importance and suitability, ecological supply and demand, ecological risk, and biodiversity conservation [38,39]. Considering the particularity of the study area, this study evaluated the importance of ecosystem service function, eco-environmental sensitivity, and landscape connectivity to form the framework of ecological source identification.

Ecological resistance surface refers to the degree of difficulty or disturbance encountered by species when moving between different landscape units or habitat patches [9]. There are three main methods applied. The first is to construct an ecological resistance surface based on the corresponding resistance coefficient of the land-use type. Although this method is simple and convenient, it is difficult to demonstrate the heterogeneity of the same type of land under different utilization states. The resistance coefficients of land-use type in different studies are additionally quite different. The resistance factor comes from in building a second to select a variety of ecological evaluation index systems, although this method more comprehensively reflects the ecological resistance. Its disadvantages are also obvious, such as that the academic has not yet unified how to assign weight to resistance factors and assign graded resistance coefficients at present. In addition, the difference between different resistance factors cannot be reflected by assigning corresponding resistance coefficients to different resistance factors [40]. The third is to improve the basic ecological resistance coefficient based on land-use type, night-time light data, moisture index, and terrain factors [41,42]. Combined with the actual situation of the study area and the availability of data, the third method was selected to construct the ecological resistance surface, which could eliminate the influence of the intensity of human activity, wetness, and topography on the ecological resistance coefficient.

An ecological corridor is a channel that transmits ecological flow, ecological process, and ecological function in a region [20], thereby effectively connecting ecosystem elements [19]. There are presently several methods to identify ecological corridors, including the minimum cumulative resistance (MCR) model, ant colony algorithm, spatial clustering method, and circuit theory [34,43,44,45,46]. Among them, compared with the traditional conceptual model and mathematical model, the MCR model better simulated the interference of each landscape unit to the spatial movement process of ecological flow. This study, therefore, chose the MCR model to identify the ecological corridor in the study area. An ecological node refers to the location that plays a key role in ecological flow transportation, exchange, and species migration and is of great significance to the construction of ecological security patterns [47]. However, scholars have not yet determined the definition, boundary, and location of ecological nodes. Referring to a large number of studies, this study divides ecological nodes into three categories—strategic point, breaking point, and temporary pause point [48].

Nujiang Prefecture has complex natural conditions, a sensitive ecological environment, and rich biodiversity. It is an ecological security barrier in southwest China. Secondly, it is also one of the areas with the most prominent contradiction between protection and development, so the it is the first choice as the study area. There were three detailed objectives: (1) to conduct a comprehensive assessment of ecological security based on the framework of “importance-sensitivity-connectivity” (Importance of ecosystem service, eco-environmental sensitivity, and landscape connectivity) and identify ecological sources; (2) to use the MCR model to identify ecological corridors and nodes to construct the ecological security pattern of Nujiang Prefecture; and (3) to put forward corresponding strategies for optimizing the ecological security pattern in Nujiang Prefecture. The innovation of this study is that, due to the geographical environment of Nujiang Prefecture, on the basis of ecosystem services, soil erosion, and landscape connectivity, geological hazard sensitivity is also added to comprehensively evaluate the ecological security and construct the ecological security pattern. The results provide suggestions for ecological conservation and developing ecological security in the alpine canyon area.

## 2. Study Area and Data Sources

### 2.1. Study Area

Nujiang Prefecture is located in the northwest of the Yunnan province and adjacent to Myanmar. Its geographical coordinates are between 98°32′ E~99°38′ E and 25°33′ N~28°23′ N, covering an area of 14,703 km^2^ (Figure 1). Nujiang Prefecture belongs to a subtropical mountain monsoon climate and is located in a unique plateau mountain environment with an altitude difference of 4390 m, which is a typical deep-cut zone of high mountains and valleys. Located in the middle and upper reaches of three international rivers (Salween, Lancang-Mekong, and Irrawaddy), the state is an important ecological security barrier area of the country, shouldering important missions and responsibilities in achieving carbon neutrality. Nujiang Prefecture is also the core area of the “Three Rivers Parallel Flow” natural heritage site, which covers an area of 6489.65 km^2^, accounting for 38.17% of the entire heritage site area and 44.13% of the territory of the prefecture. It is also rich in biological resources and diverse in the natural and cultural landscape. Due to the special topography of this area, with high mountains, deep valleys, steep slopes, and rapid water, its population is primarily distributed in alluvial fans along the “Three Rivers”. The phenomenon of steep slope reclamation is very common, resulting in serious soil erosion, frequent geological disasters, and serious damage to the ecological environment in Nujiang Prefecture.

### 2.2. Data Sources

Several datasets were used in this study. (1) A digital elevation model (DEM) was derived from ASTER GDEM data of the Geospatial Data Cloud (http://www.gscloud.cn/, (accessed on 25 January 2022)), with a spatial resolution of 30 m. (2) The vegetation net primary productivity (NPP) data were obtained from NASA (https://www.nasa.gov/, (accessed on 5 November 2021)) from 2010 to 2019, with a spatial resolution of 500 m. (3) The normalized difference vegetation index (NDVI) data were obtained from NASA (https://www.nasa.gov/, (accessed on 5 November 2021)) from 2019, with a spatial resolution of 250 m. (4) The NPP-VIIRS night-time light data were procured from the Earth Observation Group (https://eogdata.mines.edu/, (accessed on 2 December 2021)), with a spatial resolution of 500 m. (5) The administrative boundary data and road data were procured from the National Basic Geographic Information Database (https://www.ngcc.cn/, (accessed on 12 November 2021)). (6) The fault and lithology data were extracted from the 1:250,000 geological map provided by Nujiang State Land and Land Bureau. (7) The meteorological data were procured from the Nujiang Meteorological Bureau and Water Bureau. (8) The land-use type data were carried out based on the Landsat 8 OLI/TIRS data of Geospatial Data Cloud (http://www.gscloud.cn/, (accessed on 25 December 2020)) from 2020 and the verification of field sampling survey. The interpretation accuracy reached 94.17%.

## 3. Methods

This study primarily comprises three stages, as presented in Figure 2. The ecological security of Nujiang Prefecture was first comprehensively evaluated based on the importance of ecosystem services, ecological environment sensitivity, landscape connectivity, and ecological sources identified. Secondly, based on the ecological sources and ecological resistance surface, the MCR model was used to identify the ecological corridors in Nujiang Prefecture. The ecological nodes were finally identified using hydrologic analysis and the ecological corridor.

### 3.1. Comprehensive Evaluation of Ecological Security

#### 3.1.1. Importance Assessment of Ecosystem Services

The importance assessment of the ecosystem service was to evaluate the capability of typical ecosystem services in the study area according to the specific ecological environment conditions. Based on GIS and RS, selecting reasonable evaluation factors and models, spatial overlay, and division level, a high level of importance plaques for protection were put forward. Nujiang Prefecture has abundant water resources and good water quality, as well as biological resources. However, the mountainous area is widely spread, with steep slopes and severe soil erosion. Therefore, the selection of water conservation, soil conservation, carbon sequestration, and habitat quality is important for the ecosystem service function evaluation factor. Water conservation, soil conservation, and carbon sequestration are obtained by net primary productivity (NPP). A quantitative index evaluation method to assess the importance of biodiversity in habitat quality using Invest model evaluation is analyzed. The evaluation formula of each factor is presented in Table 1.

#### 3.1.2. Assessment of Ecological and Environmental Sensitivity

Eco-environmental sensitivity refers to the adaptability of an ecosystem to external pressure or disturbance and the corresponding resilience after damage [53]. A large number of studies have shown that eco-environmental sensitivity is an effective and comprehensive indicator to measure the self-regulation and resilience of an ecosystem under pressure [54,55,56]. In the long run, the declining sensitivity of the ecological environment will inevitably affect the stability of the ecosystem, which in turn will hinder human development and social progress [57].

##### Soil Erosion Sensitivity

The soil erosion in Nujiang Prefecture is dominated by hydraulic erosion. The modified General Soil Erosion Equation (RUSLE) model [58] was adopted to evaluate the sensitivity of soil erosion and calculate the amount of soil erosion in the study area. The RUSLE model has wide applicability due to its simple form, few required parameters, and quantification of soil erosion based on pixels. It has been well applied in the mountainous areas of southwest China and can guide soil and water conservation planning and soil loss control under different land cover conditions (such as farmland, pasture, and disturbed forest land). The formula used is as follows:(1)$$R=0.1833\times {\left[N^{-1}\times \overset{N}{\underset{i=1}{\sum }}\frac{\sum_{j=1}^{12}P_{i,j}^{2}}{\sum_{j=1}^{12}P_{i,j}}\right]}^{1.9957}$$
where *R* is the rainfall erosivity factor [59], the unit is [MJ·mm/(hm^2^·h·a)]; *P_i,j_* is the rainfall in month *j* of year *i* in 2001–2020, the unit is [mm]; *N* is the number of years.
(2)$$\begin{array}{ll} K & =\left\{0.2+0.3exp\left[0.0256W_{g}\left(1-\frac{W_{p}}{100}\right)\right]\right\}\times {\left[\frac{W_{p}}{W_{c}+W_{p}}\right]}^{0.3}\\ & \times \left[1-\frac{0.25W_{o}}{W_{o}+\exp(3.72-2.95W_{o}}\right]\times \left[1-\frac{0.7\beta }{\beta +exp\left(22.9\beta -5.51\right)}\right] \end{array}$$
(3)$$\beta =1-\frac{W_{g}}{100}$$
where *K* is the soil erosivity factor [60], the unit is [t·h/MJ·mm]; *W_g_*, *W_p_*, *W_c_*, and *W_o_* are the percentage contents of sand, silt, clay, and organic carbon in the soil, respectively.
(4)A=R×LS×K×C×P
where *A* is the annual average soil erosion modulus per unit area in [t/hm^2^·A], *LS* is the slope length and slope factor, *C* is the vegetation cover and management factor, and *P* is the soil and water conservation factor.

##### Susceptibility to Geological Disasters

Based on the collected data on historical disasters, this study comprehensively considered the occurrence characteristics and process of geological disasters and selected slope, aspect, elevation, rivers, formation lithology, and fault structure as the internal leading factors. NDVI, road distance, land use type, and rainfall were selected as the external environmental trigger factors. The sensitivity evaluation of geological disasters is carried out by combining the GIS and *CF* models. The calculation formula used is as follows:(5)$$CF=\{\begin{array}{c} \frac{P_{a}-P_{s}}{P_{a}\left(1-P_{s}\right)},P_{a}\geq P_{s}\\ \frac{P_{a}-P_{s}}{P_{s}\left(1-P_{a}\right)},P_{a}\leq P_{s} \end{array}$$
where *CF* is the deterministic coefficient; *P_a_* is the conditional probability of the occurrence of geological disaster events in the evaluation factor a category (level); *P_s_* is the prior probability of geological disaster events occurring in the study area *S*. The value range of *CF* is [−1,1], where *CF* > 0 indicates a high possibility of geological disasters.

#### 3.1.3. Landscape Connectivity

Landscape connectivity may be defined as the degree to which landscape patches promote or hinder the movement between existing species or other ecological flows and is an important indicator in evaluating ecological processes [61]. Nujiang Prefecture is a typical high mountain valley region with complex and changeable land-use types, where the importance of landscape connectivity is much higher than that of plain cities [62]. Based on Conefor Sensinode 2.6, the Integral index of Connectivity (*IIC*), Probability of Connectivity (*PC*), and Plaque Importance Value (*PI*) indices were selected as the analysis indices to evaluate the landscape connectivity of the study area. The formula of each landscape index is presented in Table 2.

### 3.2. Constructing Ecological Security Pattern

#### 3.2.1. Identification of Ecological Sources and Points

According to the results of the comprehensive evaluation of ecological security, the primary sources are the area of the highest level, and the secondary sources are the area of the higher level. Due to the fragmentation of most ecological sources, their number is too large, and with the increase in the minimum area threshold, the number of ecological source patches decreases rapidly. Therefore, it is necessary to set the threshold of the source patches to identify the key ecological sources and then extract the centroid of the key ecological sources, namely the ecological points by using the screening tool in ArcGIS.

#### 3.2.2. Construction of Ecological Resistance Surface

Firstly, the resistance coefficient was assigned to the data of each land-use type in Nujiang (Table 3) to construct the basic ecological resistance surface according to the impact of each land-use type on the ecological environment in Nujiang [26,63]. Considering the different degrees of human disturbance of various land-use types and typical mountain canyon landforms in Nujiang Prefecture, this paper adopts noctilucent remote sensing data and DEM and uses Formulas (3) and (4) to modify the basic ecological resistance surface to obtain the ecological resistance surface of Nujiang Prefecture.

(1)The calculation formula of the modified ecological resistance coefficient based on noctilucent remote sensing data is as follows:
(6)$$R_{i}=\frac{NL_{i}}{NL_{a}}\times R$$where *R_i_* is the resistance coefficient modified based on noctilucent data; *NL_i_* is the light index of grid *i*; *NL_a_* is the average light index of land use type a corresponding to *i* [64].(2)The calculation formula of the modified ecological resistance coefficient based on elevation data is as follows:(7)$$R_{j}=\frac{DEM_{i}}{DEM_{a}}\times R_{i}$$
where *R_j_* is the ecological resistance coefficient modified based on elevation value; *DEM_i_* is the elevation value of grid *i*; *DEM_a_* is the average elevation value of land-use type a corresponding to *i* [65].

#### 3.2.3. Identification of Ecological Corridors

Knaapen et al. proposed the minimum cumulative resistance model (MCR) in the 1990s [66]. Based on the MCR model, this study identified ecological corridors with identified ecological sources as source data and ecological resistance surface as cost data by the cost distance tool and the cost path in ArcGIS. The basic calculation formula of the MCR model is as follows:(8)$$\mathrm{MCR}=f_{min}\overset{i=m}{\underset{i=n}{\sum }}D_{ij}\times R_{i}$$
where MCR is the minimum cumulative resistance; *D_ij_* is the distance from *j* to *i*; *R_i_* is the ecological resistance coefficient of *i*; *f* represents a monotonically increasing function.

#### 3.2.4. Identification of Ecological Nodes

In this study, the ridgelines were identified using the hydrological analysis tool in ArcGIS, and the ecological corridors and their intersections were defined as the ecological nodes in the study area [67]. These were further divided into strategic points, breaking points, and temporary points. A strategic point refers to the most vulnerable place in the ecological corridor. This study extracted the “ridge line” of the ecological resistance surface with the hydrological analysis method and took its intersection with the ecological corridor as a strategic point [68]. Large transportation facilities or large river systems isolate the original complete habitat, which causes obstacles to the migration and diffusion of terrestrial organisms. Considering that there is no railway in Nujiang, the intersection of the national road, provincial road, three major rivers, and the ecological corridor in Nujiang is regarded as the ecological fracture point. The respite points would be the “stepping stones”, which may promote species migration and diffusion and improve the overall connectivity of the study area. In this paper, the intersection of the “valley line” of the ecological resistance surface and the ecological corridor was used as a respite point.

## 4. Results

### 4.1. Importance Assessment of Ecosystem Services

The results of various ecosystem service functions are presented in Figure 3. The water conservation capacity of the entire Nujiang Prefecture was found to be relatively low. The low-value areas were primarily distributed in Lanping County and southeast of Lushui City, while the high-value areas were primarily concentrated in Gongshan County and Fugong County. The high-value region of soil conservation capacity was distributed in Gongshan County, and the middle-value region was primarily distributed in Fugong County and the north of Lushui City. Lanping County and the south of Lushui City were found to have the worst soil conservation capacity. The high values of carbon sequestration capacity were widely distributed in the study area, while the low values were concentrated in the high-altitude areas and the Lantsang River. In the evaluation of the habitat quality, the low-value areas were concentrated in Lanping County, Lumadeng Township, Shangpa Town of Lushui City, and Fugong County. The low-value areas in Lanping County were the largest, and the damage to habitat quality was the most severe along the Lantsang River and in the center of Lushui City. The highest habitat quality was found in Gongshan county and northern Fugong County.

Based on the above four factors, the results of the importance of ecosystem services in Nujiang Prefecture are presented in Figure 4. The importance of ecosystem services in the whole study area primarily fell under the more important and most important areas. The area of the highest importance is the largest, accounting for 37.03%, and distributed in a patchy pattern in Gaoligong Mountain Nature Reserve and northern Fugong County. The more important areas cover an area of 4268.85 km^2^, distributed in the high-altitude area in the northern region of the Gaoligong Mountain National Nature Reserve, Biluo Snow Mountain, and Yunling Nature Reserve. The medium and less important areas are concentrated in the south of Lushui City and Lanping County. The area of least importance in Lanping County is the largest, accounting for 81.67% of the total area of least importance. It is primarily distributed on the banks of the Lantsang River, Tongdian town and Jinding Town of Lanping County, and Liuku Town of Lushui City. The less important area is primarily distributed around the least important area, with an area of 1265.52 km^2^. The area of medium importance is 2762.97 km^2^, accounting for 18.79% of the total area, distributed around the areas of less and least importance. A small part is concentrated in Gudeng township and Chenggan Township.

### 4.2. Ecological and Environmental Sensitivity Assessment

According to the grading standard of hydraulic erosion intensity in the Soil Erosion Classification and Grading Standard (SL190-2007) issued in 2008, the results of the evaluation of the soil erosion sensitivity in Nujiang Prefecture were divided into six levels—micro erosion, mild erosion, moderate erosion, serious erosion, polar erosion, and severe erosion. As presented in Figure 5a and Table 4, the area of micro-erosion is the largest, accounting for 9189.75 km^2^ and accounting for 62.50% of the total area. It is widely distributed in most areas of Nujiang Prefecture in a continuous massive form. The area of extreme intensity and severe erosion is the smallest, accounting for only 324.51 km^2^ and 2.2% of the total area, primarily distributed in areas along both sides of the Nujiang River and the Lantsang River. In terms of the mean erosion modulus and erosion amount, the total amount of soil erosion in Nujiang Prefecture is approximately 1.51 × 10^7^ t, among which, the erosion amount of mild erosion is approximately 3.41 × 10^6^ t, accounting for the largest proportion. The mean soil erosion modulus in Nujiang Prefecture is 450,594 t/(km^2^·a), indicating that Nujiang Prefecture is dominated by micro erosion and mild erosion.

The results of the sensitivity evaluation of geological disasters in the study area were divided into five levels—least sensitive, less sensitive, medium sensitive, more sensitive, and most sensitive. As presented in Figure 5b and Table 5, the more sensitive and most sensitive areas of geological disasters in Nujiang Prefectureare primarily distributed along the Dulong River, Nujiang River, Lantsang River, and several tributaries, as well as alongside roads at all levels. The more and most sensitive areas comprised 3445.29 km^2^, accounting for only 23.44% of the total area. However, there are 1346 geological disaster points accounting for 88.78% of the total geological disaster points, the density being as high as 0.3907/km^2^, thereby indicating that the higher the sensitivity to a geological disaster, the greater possibility of a geological disaster.

The evaluation results of eco-environmental sensitivity were obtained by integrating the above, as presented in Figure 5c. The ecological environment of Nujiang is characterized by a small number of more and most sensitive areas, which are distributed in discontinuous bands along both sides of Nujiang river and Lantsang River, covering an area of 1170.37 km^2^ and accounting for 7.96% of the total area. The less and least sensitive areas accounted for 27.63% and 25.95%, respectively, which were distributed in a flake pattern along the two banks of the Dulong River, on both sides of the more and most sensitive areas and along the road. The least sensitive area was found to be the largest, covering 5655.38 km^2^ and accounting for 38.46% of the total area. The least sensitive areas were widely distributed in the areas with higher elevations, including Xonlika Mountain, Gaoligong Mountain, and Biluo Snow Mountain, as well as most areas of Yunling Mountain.

### 4.3. Landscape Connectivity Evaluation

The dIIC index values of 56 ecological patches in the study area ranged from 0.044 to 62.2056, and the dPC index values ranged from 0.039 to 67.3606, indicating the connectivity importance of ecological patches in the study area changed significantly. As observed in Figure 6, the area of the low-grade patch importance index was 3700.70 km^2^, which was primarily distributed in the high-altitude ice-covered land of Gongshan County and densely populated areas along the banks of Nujiang River and Lantsang River. The area of the medium and lower levels was found to be the smallest, only accounting for 2.10% of the total area and only distributed in Gongshan County. The area of the higher level was 4393.67 km^2^, primarily distributed in the areas of Biluo Snow Mountain on the right side of Nujiang River and most areas of Lanping County. The area of the highest level was the largest, accounting for 42.85%, primarily distributed in the Gaoligong Mountain Nature Reserve, the southern section of Nujiang Grand Canyon National Park, and part of the Yunling Nature Reserve.

### 4.4. Comprehensive Evaluation of Ecological Security

The distribution of ecological security in Nujiang Prefecture is presented in Figure 7a. The level of ecological security in the whole Nujiang Prefecture varied greatly, with the western region significantly higher than the eastern, with the highest-level areas concentrated in the Gaoligong Mountain Nature Reserve. The area of a higher level was found to be the largest, accounting for 28.73% of the total area and distributed around the highest-level area, followed by sporadic distribution around the medium-level area of Lanping County. The distribution of the medium-level area was relatively scattered, with an area of 3523.31 km^2^, primarily distributed around the lower and lowest level areas, followed by the high-altitude areas of Gongshan County and Laowo Mountain. The lower and lowest level areas were relatively small, covering 2270.79 km^2^ and 1402.90 km^2^ and accounting for 15.44% and 9.54% of the total area, respectively. They were primarily distributed in the southeast of Lushui City, Jinding Town, and Tongdian Town of Lanping County and on both sides of the Lantsang River.

### 4.5. Constructing the Ecological Security Pattern

#### 4.5.1. Identification of Ecological Sources and Ecological Points

Based on the comprehensive evaluation results of ecological security in Nujiang Prefecture, the patches of highest and higher ecological security were selected as the primary and secondary ecological sources. As presented in Figure 7b, the areas of primary and secondary ecological sources in Nujiang Prefecture are 3281.35 km^2^ and 4224.64 km^2^, respectively, accounting for 22.32% and 28.73% of the total area. The distribution of the ecological source areas is not balanced in Nujiang Prefecture. The large area of the ecological sources is located in the north and west of Nujiang Prefecture. The ecological sources of Lushui city and Lanping County are small and scattered. Most of the ecological sources comprise forests, where the primary ecological sources are mostly distributed in the Gaoligong Mountain Nature Reserve, Pingma National Scenic Spot, Moon Mountain National Scenic Spot, and Laowu Mountain National Scenic Spot. The secondary ecological sources are primarily distributed in areas with rich vegetation, such as the Gongshan National Scenic Spot, part of Yunling Nature Reserve, and Laowushan National Scenic Spot.

As observed in Figure 8, there are 11 primary ecological points and 17 secondary ecological points, corresponding to 1–11 and 12–28 in the figure, respectively. Gongshan county had the most ecological points, with five primary ecological points and six secondary ecological points. The second was Lushui City, with two primary ecological points and four secondary ecological points. There were three primary ecological points and three secondary ecological points in Lanping County. Fugong County had one primary ecological point and four secondary ecological points. The distribution of ecological points in each township was found to be as follows: 1–4 and 12 in Dulong Township, 5 and 16 in Cikai Town, 6 and 19 in Lishadi Township, 10 and 25 in Tue Township, and 13–15 in Bingzhongluo Township. In terms of primary ecological points, 7–9 and 11 were located in Zhongpai Township, Shideng Township, Chengbar Township, and Daxingdi Township, respectively. In terms of secondary ecological points, 17, 18, 20–24, and 26–28 were located in Plati Township, Maji Township, Shangpa Township, Hexi Township, Tongdian Township, Gudeng Township, Laowo Bai Nationality Township, Shangjiang Township, and Liuku Township, respectively.

#### 4.5.2. Identification of Ecological Corridors and Ecological Nodes

The ecological corridors were divided into primary and secondary levels. In Figure 9a, it presents 26 primary ecological corridors with a total length of 755.40 km, 39 secondary ecological corridors with a total length of 929.26 km, and three river corridors with a total length of 550.47 km. The ecological corridors are primarily short-distance with close connections, with a few corridors in Nujiang Prefecture. The primary ecological corridor was concentrated in the west of Nujiang Prefecture, while the secondary ecological corridor was primarily concentrated in the east of Nujiang Prefecture. The river corridors are the three major rivers in the study area—the Dulong River, Nujiang River, and Lantsang River. The length of the primary and secondary ecological corridors varies greatly among counties and cities. Gongshan County has the longest primary ecological corridor of 255.79 km. Secondly, the corridors in Fugong County and Lushui City are 217.81 km and 182.34 km, respectively. Lanping County has the shortest primary ecological corridor length of 95.46 km and the longest secondary ecological corridor of 456.31 km. The lengths of the secondary ecological corridors in Gongshan County and Lushui City are 180.16 km and 223.31 km. The shortest secondary ecological corridor is 60.48 km in Fugong County.

In this study, 82 ecological nodes were identified, including 26 strategic points, 36 breaking points, and 20 pose points. As presented in Figure 9b, the strategic points are primarily distributed near the ecological corridor of Lanping County. The fault points are mainly distributed on major traffic roads and rivers such as Nujiang River, Lantsang River, National Highway 219, 215, and Provincial Highway 237.

## 5. Discussion

### 5.1. Discussion on Comprehensive Ecological Security and Ecological Corridors

The distribution of ecological security in Nujiang Prefecture is extremely unbalanced, among which the low ecological security zone is concentrated in the southeast of Lushui City and Lanping County, which is the area with the most concentrated economic development and dense population distribution in Nujiang Prefecture. The land use in this region is mainly cultivated land, and the phenomenon of steep slope planting is extremely serious, especially along the Lantsang River, where soil and water loss are serious, which can easily induce geological disasters. This is also the reason for the high ecological environment sensitivity. Farmland is an artificial ecosystem with frequent agricultural activities, which can easily destroy the balance of the ecosystem. The agricultural ecosystem structure in Nujiang Prefecture is relatively simple, which leads to the low importance of ecosystem services in this region [69]. Secondly, with economic development, urban expansion and the increased intensity of mineral mining have destroyed landscape connectivity and posed an indispensable threat to ecological security. In this area, in maintaining ecological security, it is necessary to strengthen the control of arable land on steep slopes, take preventive measures against geological disasters, develop more industries with different agricultural ecosystem structures, increase the ecological restoration of mining, and prohibit urban expansion into high ecological security areas. In the middle ecological safety zone, most of them are high-altitude areas covered by snow and ice, with low vegetation coverage and landscape connectivity, which affect the migration and expansion of species. Therefore, human activities in this area should be reduced. The high ecological safety zone is mainly forest, which is mainly distributed in various nature reserves. There is less human activity and strong ecological function. It should be kept as integral as possible in the subsequent ecological protection and landscape planning.

The difference in length between the secondary ecological corridor and the primary ecological corridor is 173.8 km. Although there are a few primary ecological corridors, they can connect the ecological sources that are distributed far away. The network distribution of the ecological corridors in Nujiang Prefecture is of great significance for the protection of biodiversity. Vegetation restoration along the corridors should be strengthened to ensure the connectivity of the ecological corridors. Lushui county and Lanping County are concentrated distribution areas of secondary ecological corridors. Due to the agglomeration of economy and population, the ecological source area is relatively fragmented and small, leading to the majority of its short-distance corridors. The secondary ecological corridors can establish an artificial forest belt, not only connecting various ecological sources and facilitating the activity and migration of species but can also serve as an anti-pollution buffer zone to reduce pollution and improve the ecological security level of densely populated areas. According to the ecological corridor length of various landscape types in the study area, the length of forests was found to be the longest. The lengths of the primary and secondary ecological corridors were 748.39 km and 919.15 km, respectively, accounting for 98.94% and 98.91% of the total length, respectively. The length of the ecological corridors in grassland, cultivated land, unutilized land, water, and construction land accounted for less than 1%. It indicates that forests are the most important part of the ecological corridor, with high ecological function. It plays a key role in ecological transport, exchange, and species migration and is suitable for the habitat and migration of terrestrial organisms. Therefore, planning and construction in the future should incorporate the building of ecological corridors with forest land as the main component, where appropriate methods should be adopted to break through the obstruction which is not good for ecological security.

### 5.2. Measures and Suggestions for Optimizing Ecological Security Pattern

To maintain ecological security, realize ecological poverty alleviation, enhance the Nujiang River, and stabilize the overall ecosystem pattern, it is necessary to adjust and optimize the composition and spatial structure of the ecological security pattern in Nujiang Prefecture and put forward optimization measures and suggestions.

Firstly, the ecological source is crucial for the survival of humans, animals, and plants in a natural system. It is the ecological security pattern of the core components. It not only plays a key role in the maintenance of regional ecological security, but also has a significant impact on ecological functions, such as biodiversity protection, regulating climate, water conservation, carbon sequestration, the release of oxygen, and food supplies. Therefore, it is necessary to effectively protect the primary ecological source areas, such as the Gaoligong Mountain Nature Reserve, Laowu Mountain National scenic spot, Yunling Nature Reserve, and other ecological patches with high ecological service functions. We should strictly implement control over the use of national land and strictly prohibit any interference from human activities or urban construction activities. We should consider the contradiction between ecological protection and economic development in many ways, and urban development and construction should be minimized in the secondary ecological source area.

Secondly, as the main channel connecting various ecological sources, the ecological corridor is of key significance to the integrity and connectivity of the regional ecosystem. Different protection measures should be taken for ecological corridors with different components. When the components of ecological corridors are artificial landscapes such as construction land and traffic facilities, it should be ensured that the ecological corridors are not occupied or blocked first, and isolation belts and artificial green spaces should be built around the corridors. When the ecological corridor is composed of cultivated land or grassland, vegetation coverage should be enhanced, ecological restoration should be carried out, and a buffer zone of appropriate width should be established. When the ecological corridor is composed of forest land, the status quo should be maintained to the greatest extent to reduce interference by human activities. The construction and protection of ecological corridors can not only improve the ecological environment of the region but also improve the overall connectivity of the research area.

Thirdly, ecological nodes should be optimized. Different types of nodes have different implications and locations. Protection and restoration measures should therefore be tailored for the different types of nodes. This not only enhances the overall connectivity of the area but also improves safety and stability. Field verification should be carried out on each ecological strategic node, and corresponding measures should be taken to restore and reclaim the ones that have been blocked and destroyed by local conditions. Ecological restoration should be strengthened, and the interference of human activities and urban expansion should be reduced to free the ecological corridors.

### 5.3. Deficiencies and Prospects

Based on GIS and RS, this study constructed and analyzed the ecological security pattern of Nujiang Prefecture, which has reference value for its ecological environment protection. However, due to the limitations of the research, certain deficiencies need to be further improved. First, due to the accessibility and quantification of data, only 10 factors such as slope, elevation, and rainfall were selected in the sensitivity evaluation of geological disasters, which may be imperfect. More evaluation indicators may be added in subsequent studies to improve the accuracy and comprehensiveness of the evaluation results. Secondly, the main factors affecting the ecological security of Nujiang Prefecture may further be analyzed, such as geological disasters, soil erosion, human activities, ecosystem services, and landscape pattern. Finally, the MCR model was adopted in this study to identify the ecological corridors in Nujiang Prefecture, where the linear corridor and its spatial location distribution were obtained without considering the setting of the corridor width, which should be further discussed in subsequent studies.

## 6. Conclusions

The study evaluated the ecological security of Nujiang Prefecture and identified ecological sources by building the “importance-sensitivity-connectivity” framework. Then, using an MCR model, it identified ecological corridors and nodes and constructed an ecological security pattern. The conclusions of this study are as follows.

(1)The importance of ecosystem services was higher in the west and lower in the east. The ecological environment of Nujiang is characterized by a small number of more and most sensitive areas which are distributed in discontinuous bands along both sides of Nujiang river and Lantsang River, and the high-sensitive areas of the ecological environment were distributed discontinuously along the banks of the Nujiang and the Lantsang River. The areas with high landscape connectivity were distributed in patches in the Gaoligong Mountain Nature Reserve and the Biluo Snow Mountain, and the areas with high landscape connectivity were primarily distributed in the high-altitude ice-covered land of Gongshan County and densely populated areas along the banks of Nujiang River and Lantsang River.(2)The overall ecological security was in a good state. The level of ecological security in the whole Nujiang Prefecture varied greatly, with the western region significantly higher than the eastern. Low ecological security areas were primarily distributed in Lanping County and the southeast region of Lushui City, and high ecological security areas were primarily distributed in nature reserve areas.(3)By constructing the ecological security pattern in Nujiang Prefecture, the total areas of primary and secondary ecological sources were 3281.35 km^2^ and 4224.64 km^2^, accounting for 22.32% and 28.73% of the total area of the prefecture, respectively. The spatial distribution of the primary and secondary ecological sources was found to be uneven and primarily distributed in nature reserves and natural scenic spots. The study identified 11 primary ecological points, 17 secondary ecological points, and 26 ecological corridors with a total length of 755.40 km, and 39 secondary ecological corridors with a total length of 929.26 km. The study additionally identified three river corridors with a total length of 550.472 km, and 82 ecological nodes comprising 26 strategic points, 36 breaking points, and 20 pose points.

The study provides a new approach to the study of ecological security in alpine canyon areas. The construction of ecological security patterns provides a reference for its ecological protection, restoration, and urban construction in Nujiang Prefecture. They can effectively serve as a significant reference to support improvements to the ecological planning of territorial space in mountainous areas.

## Figures and Tables

**Figure 1 ijerph-19-10869-f001:**
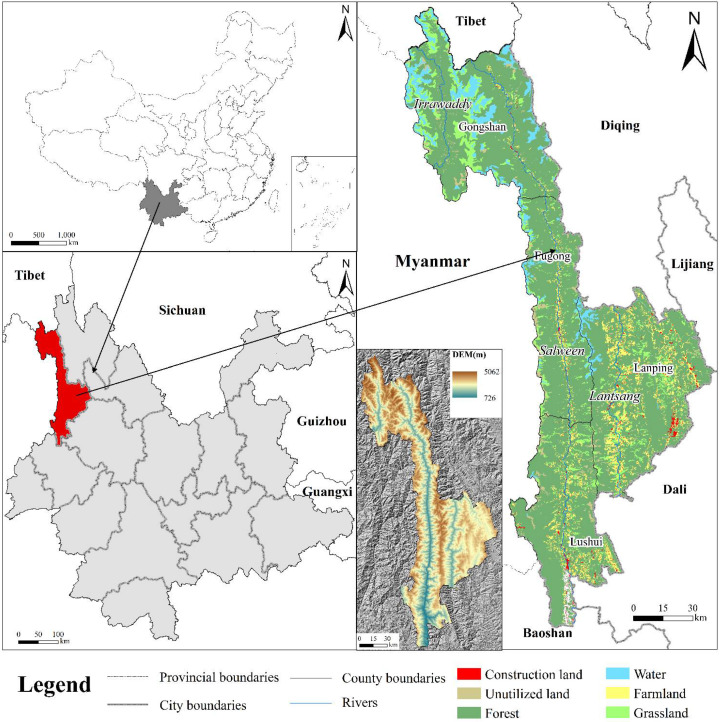
Location map of the study area.

**Figure 2 ijerph-19-10869-f002:**
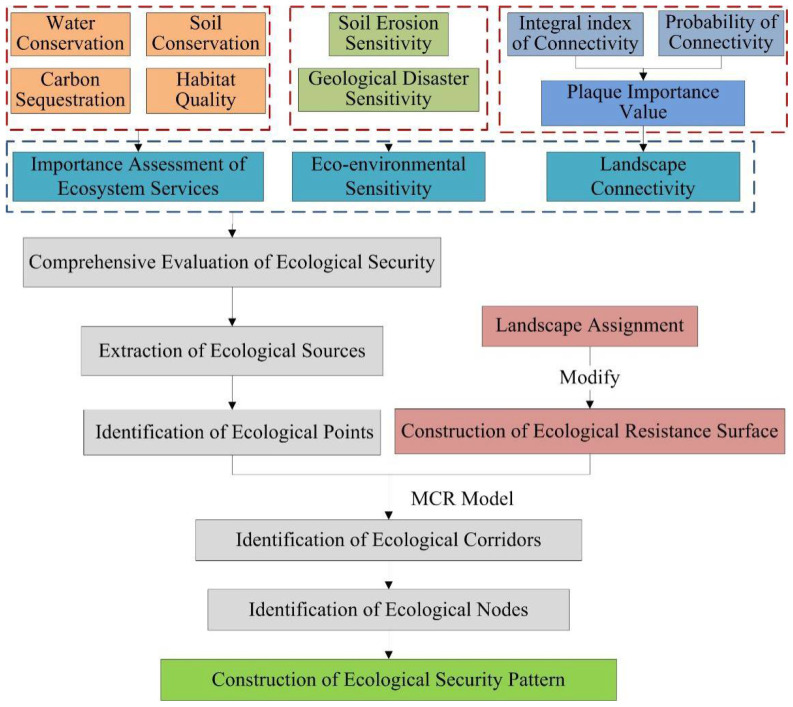
Technical route of constructing ecological security pattern.

**Figure 3 ijerph-19-10869-f003:**
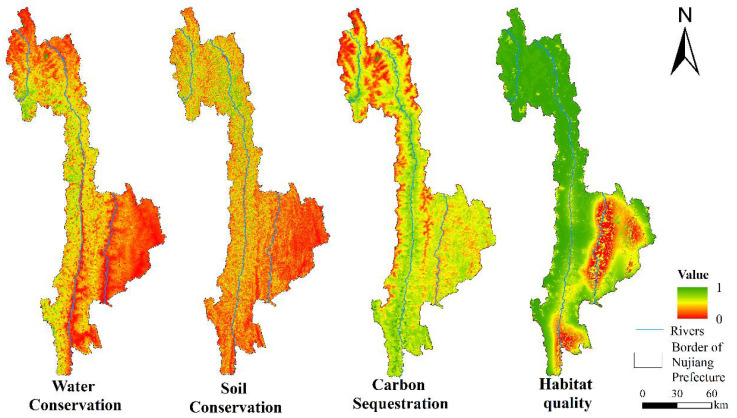
Evaluation of ecosystem service function in Nujiang Prefecture.

**Figure 4 ijerph-19-10869-f004:**
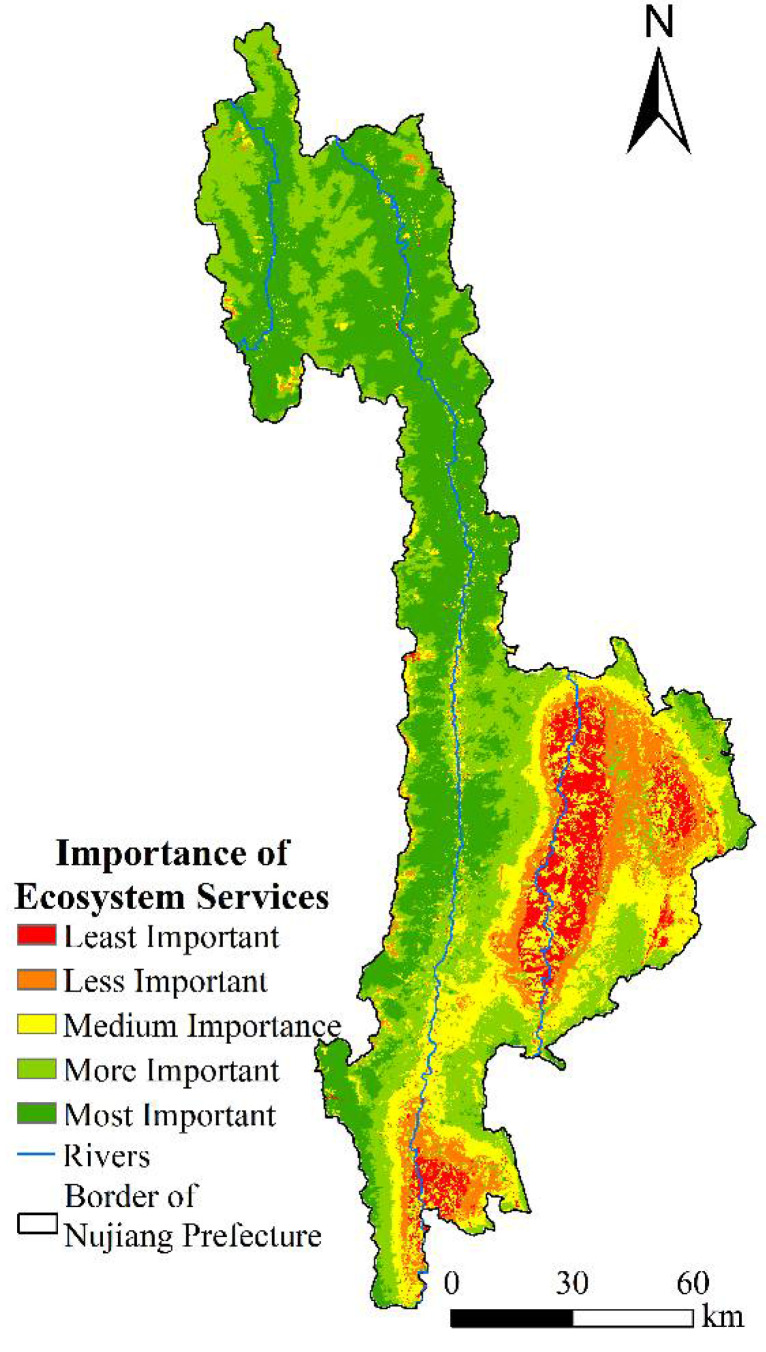
The importance of ecosystem services in Nujiang Prefecture.

**Figure 5 ijerph-19-10869-f005:**
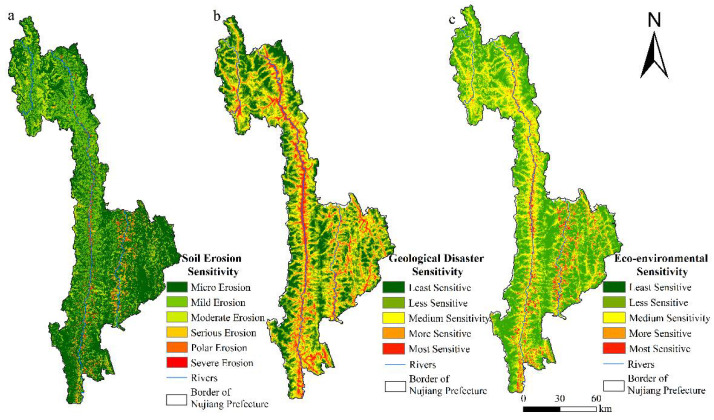
(**a**) Soil erosion sensitivity, (**b**) Geological disasters sensitivity, and (**c**) Eco-environmental sensitivity in Nujiang Prefecture.

**Figure 6 ijerph-19-10869-f006:**
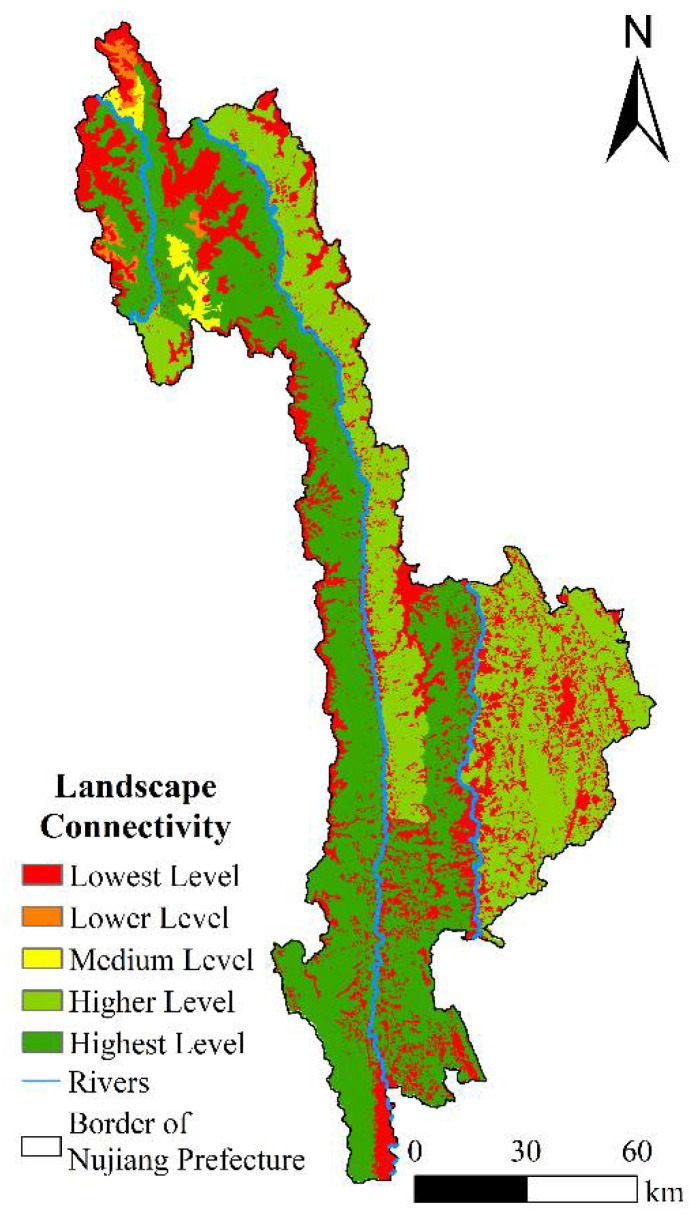
Landscape connectivity in Nujiang Prefecture.

**Figure 7 ijerph-19-10869-f007:**
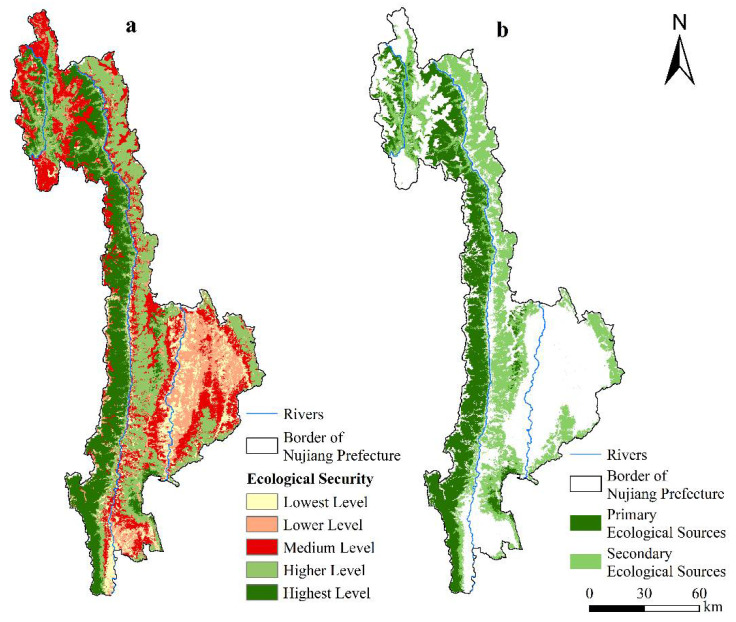
(**a**) Comprehensive evaluation of ecological security and (**b**) spatial distribution of ecological sources in Nujiang Prefecture.

**Figure 8 ijerph-19-10869-f008:**
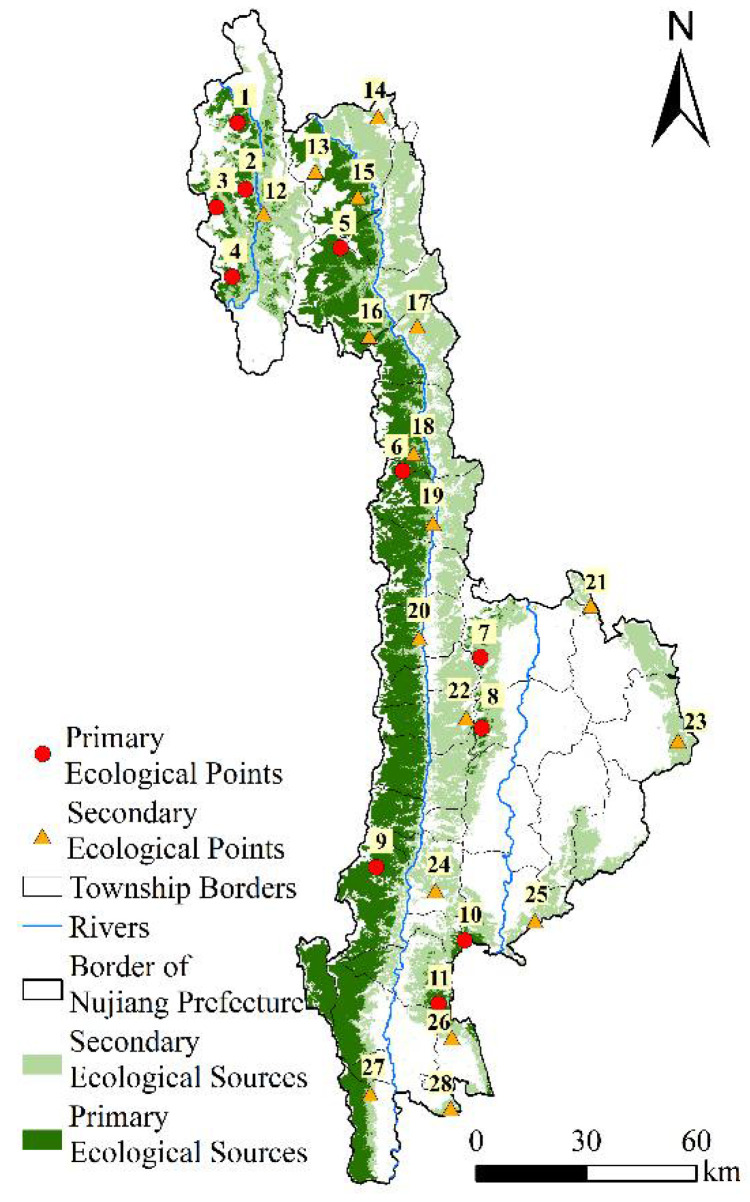
Spatial distribution of ecological points in Nujiang Prefecture.

**Figure 9 ijerph-19-10869-f009:**
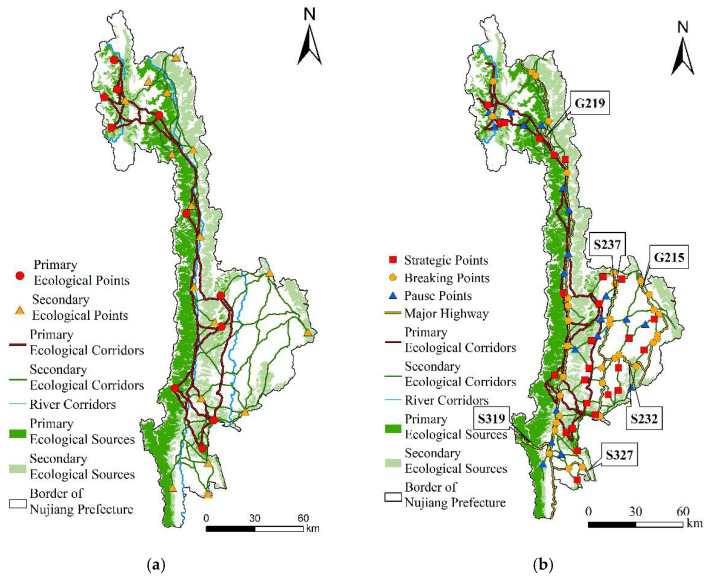
(**a**) Spatial distribution of ecological corridors; (**b**) Spatial distribution of ecological nodes in Nujiang Prefecture.

**Table 1 ijerph-19-10869-t001:** The formula of evaluation factors for the importance of ecosystem services.

Evaluation Types	Definition	Formula	Parameters Means
Water Conservation	WC refers to the ability to regulate precipitation interception, accumulation, and evapotranspiration of soil water resource [49].	$\mathrm{WC}={\mathrm{NPP}}_{\mathrm{mean}}\times F_{\mathrm{sic}}\times F_{\mathrm{pre}}\times \left(1-F_{\mathrm{slo}}\right)$	NPP_mean_ is the average annual net primary productivity of vegetation; F_sic_ is the soil infiltration factor; F_pre_ is the average annual precipitation factor, the unit is [mm]; F_slo_ is the slope factor; K is the soil erodibility factor, the unit is [t·h/MJ·mm]; 1.2 and 1.63 are constants, each 1 g dry matter can fix 1.63 g CO_2_ and release 1.2 g O_2_.
Soil Conservation	SC refers to the ability of an ecosystem to reduce or inhibit soil erosion and is a basic ecological regulation function [50].	$\mathrm{SC}={\mathrm{NPP}}_{\mathrm{mean}}\times \left(1-K\right)\times \left(1-F_{\mathrm{slo}}\right)$	
Carbon Sequestration	CS means that vegetation releases O_2_ and absorbs CO_2_ while producing organic matter through photosynthesis, which can maintain carbon and oxygen balance and regulate regional climate [51].	$\mathrm{CS}=\frac{{\mathrm{NPP}}_{\mathrm{mean}}}{45\%}\times \left(1.2+1.63\right)$	
Habitat quality	HQ refers to the ability of an ecosystem to provide suitable living conditions for individuals or populations [52].	$Q_{\mathrm{xj}}=H_{j}\left(1-\frac{D_{\mathrm{xj}}^{2}}{D_{\mathrm{xj}}^{2}+K^{2}}\right)$	Q_xj_ represents the habitat quality index of raster x in landscape pattern j; the value range of H_j_ is [0,1], representing the habitat suitability score of landscape type j; k is the half-saturation constant, which is set according to the accuracy of the data; D_xj_ is the habitat stress level of landscape type j grid x.

**Table 2 ijerph-19-10869-t002:** Landscape index formula.

Landscape Index	Formula	Parameters
Integral index of Connectivity	$IIC=\frac{\sum_{i=1}^{n}\sum_{j=1}^{n}\frac{a_{i}a_{j}}{1+nl_{ij}}}{A_{L}^{2}}$	*n* is the total number of patches; *a_i_* and *a_j_* are the areas of patches *i* and *j*, *l_ij_* is the topological distance between patches *i* and *j*; *A_L_* is the total area of the study area and is a fixed value; $p_{ij}^{*}$ is the maximum connection probability between patch *i* and patch *j*; *IIC_remove_* and *PC_remove_* are the *IIC* and *PC* values of the remaining plaques after the elimination of a single plaque.
Probability of Connectivity	$PC=\frac{\sum_{i=1}^{n}\sum_{j=1}^{n}a_{i}a_{j}p_{ij}^{*}}{A_{L}^{2}}$	
Plaque Importance Value	$dIIC=\frac{IIC-IIC_{remove}}{IIC}\times 100\%$ $dPC=\frac{PC-PC_{remove}}{PC}\times 100\%$ $PI=\frac{dIIC+dPC}{2}$	

**Table 3 ijerph-19-10869-t003:** Resistance coefficient of land use types.

Land Use Type	Forest	Grassland	Farmland	Water	Unutilized Land	Construction Land
Resistance coefficient	1	10	30	50	300	500

**Table 4 ijerph-19-10869-t004:** Soil erosion module number classification in Nujiang Prefecture.

Classification (t/km^2^·a)	Erosion Intensity	Area (km^2^)	Proportion of Total Area (%)	Mean Erosion Modulus (t/km^2^·a)	Erosion Amount (t/a)	Proportion of Total Erosion (%)
0–500	Micro erosion	9189.75	62.50	236.39	2,172,365.00	14.39
500–2500	Mild erosion	4011.77	27.29	850.61	3,412,451.68	22.60
2500–5000	Moderate erosion	783.44	5.33	3605.33	2,824,559.74	18.71
5000–8000	Serious erosion	393.53	2.68	6290.07	2,475,331.25	16.39
8000–15,000	Polar erosion	232.93	1.58	10,790.01	2,513,317.03	16.65
>15,000	Severe erosion	91.58	0.62	18,577.77	1,701,352.18	11.27
Total	-	14703	100	-	15,099,376.87	100

**Table 5 ijerph-19-10869-t005:** Sensitive classification of geological disasters and proportion of disaster sites in Nujiang Prefecture.

Sensitivity	Area (km^2^)	The Proportion of Total Area (%)	Number of Geological Hazards	Proportion (%)
Least Sensitive	3458.42	23.52%	2	0.13%
Less Sensitive	4295.31	29.21%	33	2.18%
Medium Sensitivity	3503.98	23.83%	135	8.91%
More Sensitive	2297.35	15.63%	352	23.22%
Most Sensitive	1147.94	7.81%	994	65.57%

## Data Availability

Not applicable.

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
