# Peer review of "Constructing the Ecological Security Pattern of Nujiang Prefecture Based on the Framework of “Importance–Sensitivity–Connectivity”"

_ijerph, 2022, doi:10.3390/ijerph191710869_

Round 1

Reviewer 1 Report

Based on GIS and RS technology, this paper conducts a comprehensive evaluation of ecological security and identification of ecological sources in the Nujiang Prefecture from the perspective of importance, sensitivity and connectivity. The study uses the minimum cumulative resistance (MCR) model to identify ecological corridors and ecological nodes, and optimize the ecological security structure of Nujiang Prefecture. The structure of the full text is reasonably arranged. I acknowledge the strengths of this paper, but would also like to highlight some weaknesses that I believe need to be re-addressed before publication. I hope the following comments and suggestions will help the authors to improve the manuscript and improve the quality of the paper.

The introductory part is logically reasonable, most of the cited references are from China, and few foreign examples are cited.

Materials and methods are thoroughly introduced. However, you should read carefully to try to explain more clearly each method you used in this study, although some of the language is difficult to understand and needs to be checked for revision.

The results and discussion are presented complete, but redundant, and could be written more concisely. Whether the ecological pattern under different scenarios can be simulated in the follow-up, and corresponding optimization policies are proposed.

Larger spacing between paragraphs is different from journal requirements.

Abbreviations are used for the first time without detailed explanations (similar to IIASA, IIC, PC, PI in P232).

There are problems with the application format of some references in the article, there should be spaces between the references and the article (similar to P33, P35, P53, etc.).

The names of the formulas in the third and fourth rows of Table 1 are repeated. The formats of the formulas in Table 2 are different, some formats are bold, and the formulas are not very clear; ai, aj, AL in the third column do not correspond to the formulas. Secondly, there is a problem with the format of the formula outside the table, and some parts are not marked with serial numbers.

Problems with data writing in P345 and P346: 1.51 × 107, 3.41 × 106.

Author Response

Dear reviewer:

Thank you for your precious comments and advice. Those comments are all valuable and very helpful for revising and improving our paper, as well as the important guiding significance to our research. We have studied the comments carefully and have made corrections which we hope meet with approval.

Best Regards

Yours sincerely,

ijerph-1847611

Reviewer 2 Report

Review:

Manuscript ID: International Journal of Environmental Research and Public Health - 1847611

Title: Comprehensive evaluation and pattern construction of ecological security in Nujiang Prefecture, Yunnan, SW China

Authors of the research deal with the comprehensive evaluation of ecological security and identification of ecological sources. The study identifies the ecological corridors and nodes to build ecological security patterns to optimize the ecological spatial structure of Nujiang Prefecture in China. The manuscript in an interesting and complex way presents the important topic of the study.

The aim of the study (3 detailed objectives) is formulated correctly, and the presentation of the results is satisfactory. Discussion part seems to be ok, but it could be broadened and enriched with additional references. There is recommended to add some extra references concerning the topic of ecosystem services, from different parts of the globe, e.g.:

Luty, L.; Musiał, K.; Zioło, M. The Role of Selected Ecosystem Services in Different Farming Systems in Poland Regarding the Differentiation of Agricultural Land Structure. Sustainability 2021, 13, 6673. https://doi.org/10.3390/su13126673.

Please check the instruction for authors - how to quote the references in the main text for International Journal of Environmental Research and Public Health, MDPI. Moreover, throughout the text of the manuscript there have been noted some minor grammatical errors, lack of single words, double spaces, repetitions, that should be removed or corrected. All the lines, of the manuscript should be checked, in order to remove the above-mentioned shortcomings, which were found in the main text.

Author Response

Dear reviewer:

Thanks very much for taking your time to review this manuscript. We really appreciate all your generous comments and suggestions! Please find my revisions in the re-submitted files.

Best Regards.

Yours sincerely,

ijerph-1847611

Reviewer 3 Report

Dear Authors,

Evaluation of ecological security is an subject of a great importance nowadays. This issue is characterized by high complexity and a complicated methodology.

Weaknesses of this paper:

1. A very large number of editing errors - no spaces between words, incorrect notation of units, very poor quality of the attached equations; the work is difficult to perceive due to the manner of presented information and the lack of logical transitions between individual subchapters.

2. The content of the article is not consistent with the title.

3. The aim e of the work was not clearly defined.

4. Abstract - numerous editing errors.

5. Introduction - individual paragraphs are not logically connected to each other.

6. Structure of chapter 3. Methods - requires editing;

- Table 1.very poor quality of equations,

 - failure to explain all the components of the formulas,

- same formula on Soil Conservation and Carbon Sequestration, similar situation in Table 2 Probability ...)

- redundant line numbering 199 and 213

- error in the unit - line 209

- no information on what software was used to compile the results. 

7. Results - no reference to the parameters described in 3.1.2. - 3.4.

8. Discussion - should be linked to Results; there is no discussion to be found here

9.  Titles of the Figures are not very precise.

10. Conclusions do not follow from the content of the article.

I recommend that the Authors edit the manuscript thoroughly and organize the presented content.

Author Response

Dear reviewer:

Thank you for your precious comments and advice. Those comments are all valuable and very helpful for revising and improving our paper, as well as the important guiding significance to our research. We have studied the comments carefully and have made corrections which we hope meet with approval. 

Best Regards,

Your Sincerely,

ijerph-1847611

Reviewer 4 Report

My comments for this article are as follows:

Although the intent of the work is good, the manuscript needs a revamping to elevate its quality and international readership/significance.

The topic of the article is interesting, but the manuscript is extensive, sometimes exhausting to read.

Abstract: The abstract must be improve. Must have the detailed objectives of this study.

Keywords. Must include “Minimum cumulative resistance model”.

The innovation of the article should be presented to the introduction.

Figure 2 – Where is referred “Water Conversation” will not be Water Conservation”? Where is referred “Soil Conversation” will not be “Soil Conservation”?

In Table 1, indicate the units of the Formulas. In this table, correct in the Carbon Sequestration formula where is SC = must be CS =.

Line 208 refers to the Soil erosion Equation (RUSLE). In the text you must indicate the units of the parameters of the equation.

What were the values used for the K parameter and what was the source? As we know, not all rainfall is erosive, so you should indicate which R values are used, the source and the time period.

Line 232 – what is the meaning of IIC, PC, PI ? The first time an acronym is placed in the text, it must be written in full. For example: integral index of connectivity (IIC),…

In the equations that are presented in the text, the units of the various parameters must be indicated.

Table 2 – there is repetition of Formulas

Table 3 - Resistance coefficient of land use types - indicate the source of the coefficients

Chapter 4. - This part is not well balanced. This chapter needs to be improved. The results should be written objectively.

Must be explain What were the criteria selected for the results of the importance of ecosystem services presented in Figure 4.

Table 4 – What is the source for the classes referred to in the Classification column (t/km2). Explain how the values presented in the columns “Mean erosion modulus” and “Erosion amount” were obtained.

Line 378 – what is “dIIC index”

Discussion - This chapter has to discussion the results of the study in relation to the objectives mentioned in lines 114-120, namely: “(1)to conduct a comprehensive assessment of ecological security based on the framework of “importance-sensitivity-connectivity” and identify ecological sources; (2)to construct ecological resistance considering the topography and human disturbance factors based on the land use type and corrected by noctilucent remote sensing and elevation data; and(3)to use the MCR model to identify ecological corridors and nodes to construct the ecological security pattern of Nujiang Prefecture. “

The conclusions should not be so summary and must be improved. The conclusions must be thoroughly supported by the results presented in the article or referenced in secondary literature.

References – the bibliographic references are not correct. All authors must be indicated and not only the first author followed by et al. For examples: Line 606 Kummu, M., et al. ; Line 611 He, C., et al.,…)

Author Response

Dear reviewer:

Thank you for your precious comments and advice. Those comments are all valuable and very helpful for revising and improving our paper, as well as the important guiding significance to our research. We have studied the comments carefully and have made corrections which we hope meet with approval.

Best regards,

Your Sincerely,

ijerph-1847611

Round 2

Reviewer 3 Report

1. New version of the paper still contains a very large number of editing errors - no spaces between words, error in the title - line 390; and the lack of logical transitions between individual subchapters.

2. The content of the article is not consistent with the title. It is still the same.

3. Structure of chapter 3. Methods - requires editing;

Table 1 - add units of the parameters used n the formulas; are you sure of the CS equation?

Table 2 - correct equations and remove doubled PIV formulas

4. Discussion - should be linked to Results; there is no discussion to be found here; why the first version of subchapter 5.1 was removed?

Author Response

(The authors gave the same response as above.)

Reviewer 4 Report

The article continues with problems. The authors did not meet all the improvement needs. Some of my comments remain, namely:

Abstract: The abstract must be improve. Must have the detailed objectives of this study. The aim of the work was not clearly defined.

The innovation of the article should be presented to the introduction.

What were the values used for the K parameter and what was the source? As we know, not all rainfall is erosive, so you should indicate which R values are used, the source and the time period.

Table 4 – What is the source for the classes referred to in the Classification column (t/km2). Explain how the values presented in the columns “Mean erosion modulus” and “Erosion amount” were obtained.

Line 264 - equation numbering is missing.

Table 2.very poor quality of equations

Author Response

(The authors gave the same response as above.)
